# Search for and Identification of Young Compact Galactic Supernova Remnants Using THOR

**Sujith Ranasinghe ***, **Denis Leahy** **and Jeroen Stil**

Department of Physics and Astronomy, University of Calgary, Calgary, AB T2N 1N4, Canada; leahy@ucalgary.ca (D.L.); jstil@ucalgary.ca (J.S.)
* Correspondence: syranasi@ucalgary.ca

**Abstract:** Young Supernova remnants (SNRs) with smaller angular sizes are likely missing from existing radio SNR catalogues, caused by observational constraints and selection effects. In order to find new compact radio SNR candidates, we searched the high angular resolution ($25''$) THOR radio survey of the first quadrant of the galaxy. We selected sources with non-thermal radio spectra. HI absorption spectra and channel maps were used to identify which sources are galactic and to estimate their distances. Two new compact SNRs were found: G31.299-0.493 and G18.760-0.072, of which the latter was a previously suggested SNR candidate. The distances to these SNRs are $5.0 \pm 0.3$ kpc and $4.7 \pm 0.2$ kpc, respectively. Based on the SN rate in the galaxy or on the statistics of known SNRs, we estimate that there are 15–20 not-yet detected compact SNRs in the galaxy and that the THOR survey area should contain three or four. Our detection of two SNRs (half the expected number) is consistent with the THOR sensitivity limit compared with the distribution of integrated flux densities of SNRs.

**Keywords:** supernova remnants; radio continuum; radio lines





## 1. Introduction

For a better understanding of the interstellar medium (ISM) and the evolution of our galaxy, we need not only to study the properties of individual supernova remnants (SNRs) but also to obtain an as complete as possible sample of SNRs. The current observed number of galactic SNRs is 294 [1]. There is an apparent deficit between the number of observed and expected galactic radio SNRs [2,3].

From the [1] catalogue, the SNR with the smallest angular radius has a radius $45''$. SNR G1.9+0.3 is one of the smallest galactic SNRs, and [4] finds that it is near the galactic center. With the distance to the galactic center $R_0 = 8.34 \pm 0.16$ kpc [5], the radius is 1.8 pc. Many other small SNRs with known distances have radii of $\sim 2$ pc.

Assuming a mean lifetime of $\sim$60,000 yr for radio SNRs [6] and a supernova (SN) rate of one per $40 \pm 10$ yr [7], the expected number of SNRs is $\sim$1500. The reasons for the lack of observed SNRs is likely the result of selection effects acting against the discovery of old, faint, large remnants as well as very young, small remnants due to poor sensitivity and spatial resolution [3,8]. The identification of faint, old, and large remnants remains a difficult task because of their low surface brightness and the complex structure of background radio emission. However, recent observations from the low-frequency GLEAM survey have produced a number of SNRs and candidates [9,10]. The majority of these SNRs and SNR candidates have radii ranging from $\sim 2'$ to $\sim 50'$. The SNR candidates presented by [11] using the MAGPIS data have radii of a few arcmin. In this work, we searched for compact SNR candidates (radii < few arcmin). With a $25''$ angular resolution [12] and sensitive up to $120''$ [13], the THOR survey should be ideal for identifying compact remnants.

For catalogued SNRs with distance estimates (Green's Galactic SNR catalogue [1]), $\sim$9 have a radius <3 pc and $\sim$20 have radius <5 pc. These correspond to angular radii of $39''$ and $64''$ at 16 kpc, respectively, or $75''$ and $123''$ at a more common distance of

the galactic centre (8.34 kpc). We assume for an order of magnitude estimate that the true SNR distribution with angular radius is similar to the observed distribution except for scaling. Then, we expect to observe 20 (the observed number with radius $< 2.5'$) $\times >1000/294$ (the lower limit of the scaling factor) $\times 1/5.3$ (the fraction of the total galactic SNR population covered by THOR).[1] This yields $> 12$ SNRs with an angular diameter $\lesssim 2'$ in the region covered by the THOR survey.[2] The expected number of previously undiscovered SNRs less than 1000 yr old is $\sim$15 to 20, based on the both statistics of SNRs and galactic SN rate (see Section 4.3 below).

To identify SNRs, Ref. [14] presented the following criteria:

1. Source has a shell-like morphology,
2. Source has a non-thermal spectral index (negative $\alpha$ ($< -0.2$), where $_\nu \propto \nu^\alpha$),
3. Source has weak or absent mid-infrared (MIR) emission (e.g., 8 μm, 24 μm).

For compact objects (less than several beam areas), a shell-like morphology for an SNR cannot be detected. Hence, for this work, we use the two latter criteria in order to identify SNR candidates.

An outline of the current work is as follows. In Section 2, we present a brief description of the data and method used for identifying galactic SNR candidates. The detailed procedure used to identify the candidates is given in Section 2.2. The results are given in Section 3, the discussion is given in Section 4, and the conclusion is given in Section 5.

## 2. Data and Analysis

### 2.1. Data

For the identification of SNR candidates, we used the THOR survey [12]. The survey covers a region between galactic longitudes 14.5° and 67.4° and latitudes −1.25° and 1.25° with continuum angular resolution of 25″, HI data angular resolution of 40″, and spectral resolution of 1.5 km s$^{-1}$ . For this work, we use the THOR data release 2 [15], which has the continuum spectral windows centred at 1060, 1310, 1440, 1690, 1820, and 1950 MHz.

To obtain flux densities at low frequency, we utilize the 150 MHz TIFR GMRT Sky Survey (TGSS; [16]). The TGSS data covers a region between −50° and +90° declination (Dec) with a resolution of 25″ × 25″ for Dec > 19° and 25″ × 25″/ cos(Dec − 19°) for Dec < 19° .

Molecular cloud (MC) associations were investigated using the $^{13}$CO, $J = 1 \rightarrow 0$ spectral line data (110.2 GHz) from the Galactic Ring Survey of the Five College Radio Astronomical Observatory 14 m telescope (FCRAO; [17]). The $^{13}$CO data cover a region between longitudes 18° and 55.7° and latitudes between −1° and 1° with an angular resolution of 46″ and a spectral resolution of 0.21 km s$^{-1}$. The survey's velocity coverage is −5 to +135 km s$^{-1}$ for galactic longitudes $l \leqslant 40°$ and −5 to +85 km s$^{-1}$ for galactic longitudes $l > 40°$.

To identify extragalactic sources, 5 GHz data from the Co-Ordinated Radio 'N' Infra-red Survey for High-mass star formation (CORNISH, [18]) and 1.4 GHz data from the Multi-Array Galactic Plane Imaging Survey (MAGPIS,[3] [11,19]) were used. With a resolution of 1.5″ and an rms noise level < 0.4 mJy/beam, CORNISH resolves most extragalactic objects well. The MAGPIS data cover the region between galactic longitudes of 5° and 48.5° and longitudes between −0.8° and 0.8° with an angular resolution of 6″. Reference [18] chose a $7\sigma$ ($\sim$ 2.5 mJy/beam) detection threshold for their sources. Whereas the sensitivity to extended emission of the MAGPIS data is better than of CORNISH, it has a lower resolution (6″).

To test for presence of thermal emission, we use the Galactic Legacy Infrared Mid-Plane Survey Extraordinaire (GLIMPSE; [20]) 8.0 μm data and the Multiband Infrared Photometer for Spitzer (MIPSGAL; [21]) 24 μm data. GLIMPSE covers a region between galactic longitudes 10° and 65° both sides of the galactic center and latitudes between −1° and 1° with a pixel size of 1.2″ and sensitivity of 0.4 mJy. The MIPSGAL survey covers the same area as GLIMPSE with an angular resolution of 6″.

### 2.2. Identification of SNR Candidates

#### 2.2.1. Identification of Non-Thermal Sources Which Are Not Parts of Known SNRs

To search for compact SNR Candidates, we utilize the continuum source catalogue of [22], which lists 10,387 radio sources. It contains spectral indices obtained from comparing peak intensities in the spectral windows of THOR. For non-thermal emission, typically the spectral index is $<-0.2$. There are 8230 radio sources with defined spectral indices. We first chose objects that are bright (integrated flux > 50 mJy) and have non-thermal spectral index, defined here as $\alpha < -0.2$. Bright sources were chosen to construct HI spectra, so that the source absorption features would be distinguishable from background [23].

The number of objects that meet the above criteria is 508. Of these objects, 158 were found to be part of known SNRs and thus were excluded from further analysis, leaving 350 objects.

#### 2.2.2. Identification of Extragalactic Sources

The 350 non-thermal objects were analyzed to identify extragalactic sources in two independent ways. First, we found 77 radio sources that have a double-lobe morphology in THOR. The 77 radio sources were further inspected for double lobe-morphology using MAGPIS images. Those were identified as extragalactic, leaving 273 possible galactic objects of interest.

Next, we analyzed the HI spectra of all 350 objects to independently identify extragalactic objects. The HI absorption spectrum of an extragalactic source shows absorption features at all velocities of galactic HI. The 350 spectra and channel maps were extracted from THOR data and analyzed to determine whether absorption is present in the whole negative velocity range.[4] We found that 150 of the 273 objects with no double-lobe morphology show absorption in the whole negative velocity range and are therefore extragalactic.

For the double-lobe morphology objects, we compare the HI absorption spectra of the two lobes. If they are not similar, we treat them as two separate objects with coincidental spatial association, thus removing that object from the double-lobe classification. The spectrum of G49.210-0.963 is shown in Figure 1 as an example where the absorption spectra of the two lobes agree. The two lobes have slightly different brightnesses that account for the scaling difference in the spectra and have similar optical depths. The absorption is seen in the whole negative velocity range, confirming its extragalactic nature.

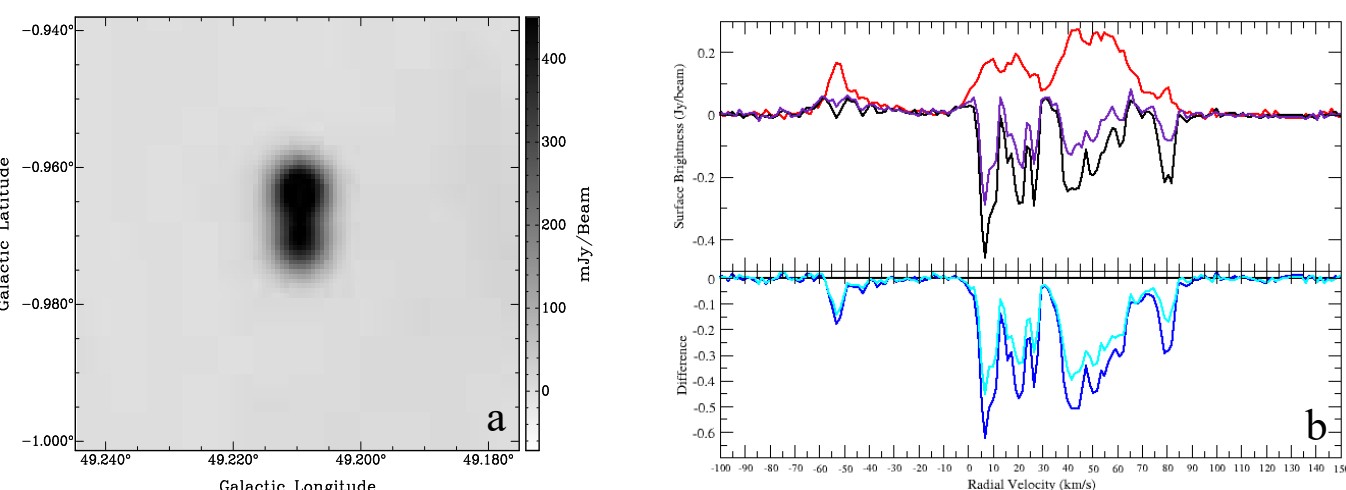

**Figure 1.** (**a**) THOR 1440 MHz continuum image of the extragalactic source G49.210-0.963; (**b**) HI spectrum: top panel: HI emission spectra (source: black (northern lobe), purple (southern lobe) and background: red); bottom panel: source—background (difference: blue (northern lobe), cyan (southern lobe)).

### 2.2.3. Selection of SNR Candidates

After the removal of identified extragalactic objects, 123 objects remain. Of these, 75 had too much noise to show clear HI absorption features. Those 75 are identified as ambiguous galactic or extragalactic and omitted from the list of verified galactic objects.

This leaves 48 verified non-thermal galactic objects, with no absorption features at negative velocities.[5] A cross-match with known HII regions [24–28] identified 31 as HII regions. The spectral indices of a majority of these HII regions have relatively large errors ($\sim$25% to $\sim$50%); thus, many are consistent with a thermal spectrum within 2 to 3$\sigma$. There is no detectable emission at 150 MHz for any of these HII regions, which indicates that they likely have thermal radio spectra.

This leaves 17 non-thermal galactic objects that remain as SNR candidates. One of these sources (G18.760-0.072) was previously identified by [29] as a SNR candidate.

After the SNR candidates were found, the following steps were taken to further check whether they are SNR candidates.

1. The SPIDX progressive survey (http://tgssadr.strw.leidenuniv.nl/hips_spidx/, accessed on 3 March 2020; [30]) was used to verify the spectral index.
2. From the GLIMPSE and MIPSGAL surveys, 5.8, 8.0, and 24 μm data were obtained to verify the weak or absent mid-infrared (MIR) emission of the candidates.
3. CORNISH and MAGPIS survey images were examined to verify the extended nature of the radio sources.
4. Angular size of the candidates was measured.
5. Using the TGSS, THOR, and CORNISH surveys, we independently obtained the spectral indices for the candidates.
6. The $^{13}$CO data were examined to determine any molecular cloud associations.

No detectable MIR emission was found in any of the candidates.

### 3. Results

In summary, we searched the list of 10387 radio sources from [22] to find 17 SNR candidates. The steps followed to obtain this list of 17 SNR candidates (non-thermal galactic sources) are summarized in Table 1.

**Table 1.** Source numbers by category.

| Category | Number of Sources in Category | Remaining Number of Sources |
|---|---|---|
| Initial THOR source catalog | 10,387 | 10,387 |
| Sources with undefined spectral indices | 2157 | 8230 |
| Sources with $S_{int} < 50$ mJy or $\alpha > -0.2$ | 7722 | 508 |
| Part of known SNRs | 158 | 350 |
| Sources with lobe-like morphology | 77 | 273 |
| Extragalactic sources (from HI spectra) | 150 | 123 |
| Sources too faint for reliable HI spectra | 75 | 48 |
| HII regions | 31 | 17 |
| SNR Candidates [a] | 17 | |

[a] 2 objects are SNRs. The rest are too compact to be SNRs (see Section 4).

### 3.1. Extragalactic Sources

As part of the procedure of searching for SNR candidates, we identified extragalactic sources, as described in Section 2.2.2 above. We present a sample of the list of extragalactic sources found in the THOR survey area in Table 2, with the full table of sources online (see Supplementary Materials).

**Table 2.** Extragalactic sources.

| # | Object | l (deg) | b (deg) | RA (deg) | Dec (deg) | Spectral Index [a] | Error | CORNISH [b] | $V_r$ [c] (km s$^{-1}$) | Double-Lobed [d] |
|---|--------|---------|---------|----------|-----------|---------------|-------|-------------|----------------------|------------------|
| 1 | G17.910+0.372 | 17.91 | 0.372 | 275.545 | −13.162 | −0.77 | 0.01 | Ex G | −35 | N |
| 2 | G17.913−0.329 | 17.913 | −0.329 | 276.184 | −13.488 | −0.99 | 0.04 | IRQ G | ... | P |
| 3 | G18.092+1.167 | 18.092 | 1.167 | 274.915 | −12.627 | −0.98 | 0.01 | Ex P | −39 | N |
| 4 | G18.106+0.186 | 18.106 | 0.186 | 275.809 | −13.077 | −0.94 | 0.01 | Ex G | −33.5 | N |
| 5 | G18.312+1.044 | 18.312 | 1.044 | 275.131 | −12.491 | −1.05 | 0.02 | IRQ G | −33.5 | S |
| 6 | G18.368−1.126 | 18.368 | −1.126 | 277.126 | −13.457 | −1.01 | 0.03 | ... | −23 | S |
| 7 | G18.495+0.273 | 18.495 | 0.273 | 275.917 | −12.692 | −1.22 | 0.03 | IR Q | −32 | N |
| 8 | G18.696−0.401 | 18.696 | −0.401 | 276.625 | −12.829 | −0.90 | 0.02 | IR Q | −32 | N |
| 9 | G18.755−0.497 | 18.755 | −0.497 | 276.739 | −12.822 | −0.77 | 0.03 | Ex G | −27.5 | N |
| 10 | G18.791+0.628 | 18.791 | 0.628 | 275.737 | −12.264 | −1.21 | 0.04 | EX G | −24.5 | S |

[a] The spectral indices are from [22]. [b] The CORNISH catalogue source type—IRQ: IR-Quiet, Ex: extragalactic; and RS: Radio-star. Each with source type confidence—G: good, M: moderate, and P: poor. [c] ... indicates that the HI absorption features are not easily identifiable. [d] P: Double-lobed structure in THOR is the primary (P) indicator of its extragalactic nature; S: Double-lobed structure in THOR is the secondary (S) indicator of its extragalactic nature; and N: source has no (N) double-lobed structure in THOR.

### 3.2. Angular Sizes of the 17 Candidates/Galactic Non-Thermal Sources

The angular sizes of the 17 SNR candidates were measured using the THOR images and verified using MAGPIS and CORNISH images, when those were available. Figure 2 shows the images of G42.028-0.605, as an example of a source that appears unresolved by THOR but is clearly seen as two compact sources in the MAGPIS and CORNISH images. The MAGPIS and CORNISH images show that G41.889-0.386 is also a pair two compact radio sources (for a discussion of the nature of these non-thermal sources, see Section 4.2).

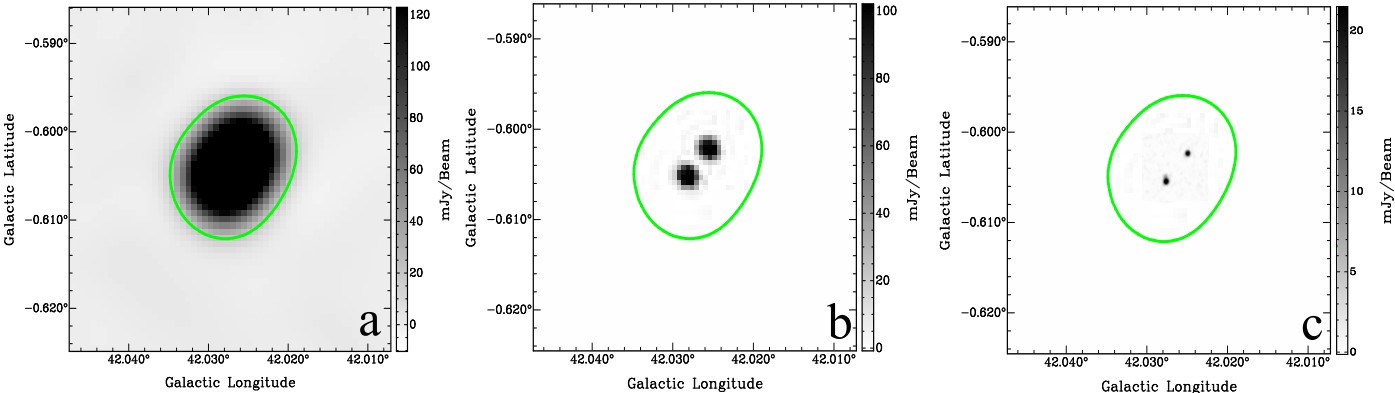

**Figure 2.** The non-thermal object G42.028-0.605. (**a**) THOR 1440 MHz; (**b**) MAGPIS 1.4 GHz; (**c**) CORNISH 5 GHz. The green contour line is at 0.02 Jy/beam of the THOR 1440 MHz image.

Table 3 gives the results of radius measurements for 8 of the 17 sources. The nine sources not listed are all unresolved, so have identical entries, except for Object. Of the 17 sources, 4 are resolved and the remaining 13 are unresolved (such as sources 4, 5, 6, and 8). Only two of the sources are larger than 1″, and many are less than 0.25″. At 8 kpc distance, 1″ (0.25″) yields a radius of 0.04 pc (0.01 pc), which for a typical young SN expansion speed of 8000 km s$^{-1}$ yields an age of 4.8 yr (1.2 yr). Given the SN rate of the galaxy of ∼1/40 yr, it is unlikely that any of the unresolved objects or the two smallest resolved ones are SNRs. We re-label these as galactic non-thermal sources (17 galactic non-thermal sources, of which the two largest remain SNR candidates) .

This leaves the two largest angular radius ones (G18.760-0.072 and G31.299-0.493) in Table 3 as SNR candidates. As these two sources have passed all of the criteria for SNRs in Section 1 and further verification tests are listed in Section 2.2.3, hereafter, we label these as SNRs.

**Table 3.** Angular radii of the galactic non-thermal sources [a].

| # | Object | $R_{THOR}$ (″) | $R_{MAGPIS}$ [b] (″) | $R_{CORNISH}$ [b] (″) |
|---|--------|----------------|----------------------|------------------------|
| 1 | G18.760−0.072 | PR [c] | 50 | ... |
| 2 | G31.299−0.493 | $\lesssim$12.5 | 10 | ... |
| 3 | G27.920+0.977 | $\lesssim$12.5 | $\lesssim$3 | PR: 0.52 |
| 4 | G35.351+0.239 | $\lesssim$12.5 | $\lesssim$3 | $\lesssim$0.75 |
| 5 | G35.484+0.424 | $\lesssim$12.5 | $\lesssim$3 | $\lesssim$0.75 |
| 6 | G41.889−0.386 | PR | $\lesssim$3/$\lesssim$3 [d] | ... |
| 7 | G42.028−0.605 | PR | $\lesssim$3/$\lesssim$3 [d] | PR: 0.8/0.74 [d] |
| 8 | G42.093−0.430 | $\lesssim$12.5 | $\lesssim$ 3 | $\lesssim$0.75 |

[a] The nine sources not listed are all unresolved, as are sources 4, 5, 6, and 8, which have identical entries. For unresolved sources, we estimate an upper limit to the angular radius of $\lesssim$ 1/2 of the telescope beam FWHM. For the THOR, MAGPIS, and CORNISH surveys, the beam FWHMs are 25″, 6″, and 1.5″, respectively. [b] If a source is resolved or partially resolved, the size is corrected for the size of the telescope beam of the highest resolution survey available. [c] PR: Partially resolved. [d] Source seen in THOR is resolved into two sources by MAGPIS or CORNISH.

### 3.3. HI Absorption Distances to the 17 Galactic Non-Thermal Sources

The distances are estimated by analyzing the HI absorption spectra to determine the lower and upper limits to the kinematic distance of each SNR candidate. The distance estimate method is described by [23], and the error analysis for the distances is described by [31]. The list of HI distances for the 17 galactic non-thermal sources is given in Table 4.

**Table 4.** Non-thermal objects and their distances.

| # | Object | 1.4 GHz Integrated Flux Density (Jy) | Spectral Index [a] | Spectral Index [b] | $V_r$ (km s$^{-1}$) | KDAR [c] | Distance (kpc) | Radius (pc) |
|---|--------|------|------|------|------|------|------|------|
| | SNRs | | | | | | | |
| 1 | G18.760−0.072 | 0.155 | −0.35 ± 0.08 | −0.52 ± 0.03 | 70 | N | 4.7 ± 0.2 | 1.2 |
| 2 | G31.299−0.493 | 0.052 | −0.65 ± 0.08 | −0.89 ± 0.04 | 85 | N | 5.0 ± 0.3 | 0.25 |
| | Sample 1 [d] | | | | | | | |
| 1 | G27.920+0.977 | 0.699 | −0.73 ± 0.01 | −0.74 ± 0.01 | 55 | F | 11.3 ± 0.3 | 0.028 |
| 2 | G35.351+0.239 | 0.071 | −0.29 ± 0.03 | −0.30 ± 0.04 −0.15 ± 0.02 | 80.5 | N | 4.9 ± 0.4 | 0.012 |
| 3 | G35.484+0.424 | 0.096 | −0.53 ± 0.02 | −0.33 \ −0.50 ± 0.03 | 87.5 | F | 6.8–13.5 | 0.016–0.032 |
| 4 | G41.889−0.386 | 0.078 | −1.06 ± 0.03 | −0.31 \ −0.89 ± 0.08 | 55 | F | 8.9 ± 0.4 | 0.035/0.022 |
| 5 | G42.028−0.605 | 0.473 | −1.01 ± 0.01 | −0.49 \ −1.09 ± 0.04 | 20.5 | F | 11.7 ± 1.1 | 0.045/0.042 |
| 6 | G42.093−0.430 | 0.078 | −0.40 ± 0.03 | −0.22 \ −0.41 ± 0.02 | 65.5 | N | 4.3 ± 0.5 | 0.010 |
| | Sample 2 [d] | | | | | | | |
| 1 | G23.088+0.224 | 0.888 | −0.95 ± 0.03 | −0.62 \ −0.93 ± 0.09 | 110.5 | TP [e] | > 7.6 | 0.018 |
| 2 | G24.430−1.017 | 0.126 | −0.44 ± 0.03 | −0.18 \ −0.43 ± 0.05 | 110.5 | TP [e] | > 7.6 | 0.018 |
| 3 | G34.421−1.031 | 0.145 | −0.29 ± 0.03 | −0.69 \ −0.36 ± 0.12 | 85 | N | 5.1 ± 0.4 | 0.017 |
| 4 | G36.204−0.342 | 0.084 | −0.73 ± 0.03 | −0.65 ± 0.02 | 92 | TP [e] | > 6.9 | 0.017 |
| 5 | G43.030−0.077 | 0.102 | −0.78 ± 0.03 | −0.37 \ −0.84 ± 0.23 | 37 | F | 9.8 ± 1.4 | 0.024 |
| 6 | G44.324−0.730 | 0.111 | −1.03 ± 0.03 | −0.96 ± 0.02 | 71.5 | TP [e] | > 6.1 | 0.015 |
| 7 | G56.364+0.617 | 0.056 | −0.67 ± 0.03 | −0.30 \ −0.56 ± 0.04 | 41.5 | TP [e] | > 4.6 | 0.011 |
| 8 | G56.608−1.105 | 0.104 | −0.70 ± 0.03 | −0.80 \ −0.58 ± 0.04 [f] | 43 | TP [e] | > 4.6 | ... [g] |
| 9 | G64.019−0.846 | 0.232 | −0.90 ± 0.03 | −0.93 ± 0.01 \ −0.26 [h] | 26.5 | TP [e] | > 3.6 | 0.009 |

[a] These spectral index estimates are from [22]. [b] The spectral indices from this work. Two values given are for the broken power law, where the first value is between the 150 Mhz to 1.06 GHz and the second value is from 1.06 to 5 GHz. [c] KDAR (kinematic distance ambiguity resolution), indicating whether the non-thermal object is at the near (N), far (F), or tangent point (TP) distance. [d] SNRs and Sample 1: Sources with nearby sources with comparison HI spectra (i.e., more reliable HI spectra). Sample 2: Sources without nearby sources with comparison HI spectra. [e] Indicates a lower limit of the distance. [f] This source has no CORNISH data. [g] This source has no CORNISH or MAGPIS data to determine the radius. [h] For this source, the first value of spectral index is for 150 MHz to 1.95 GHz. The second value of spectral index is for 1.95 GHZ to 5 GHz.

The two SNRs and the set labeled Sample 1 in Table 4 have more reliable HI distances because each of those sources have a nearby comparison source with an HI absorption spectrum, which shows absorption features that serve to verify the distances. The nine

sources that are listed as Sample 2 in Table 4 do not have nearby comparison sources with HI spectra. A detailed analyses of Sample 1 and 2 sources are given in the Appendix A.

We give details of the distance estimation for the two SNRs next.

### 3.3.1. SNR G18.760−0.072

Ref. [29] presented G18.7-0.07 as a candidate based on THOR and MIR data from the GLIMPSE, MIPSGAL, and WISE surveys. G18.760-0.072 (the object name from [22] catalogue) is a compact radio source and is 1.4′ (1440 MHz, THOR) in size. The MAGPIS 1.4 data (Figure 3b) shows an extended emission matching the THOR 1440 MHz morphology well (Figure 3a).

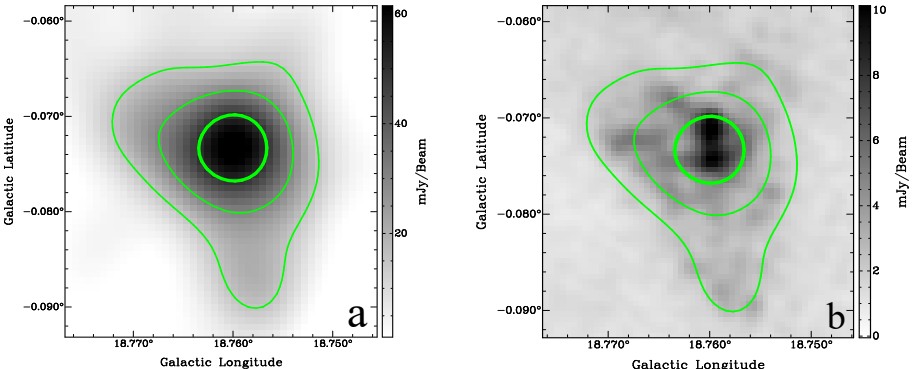

**Figure 3.** SNR G18.760-0.072 (**a**) THOR 1440 MHz; (**b**) MAGPIS 1.4 GHz image. The green contours are from the 1440 MHz THOR data at 15, 30, and 60 mJy/beam.

The HI spectrum does not show absorption up to the tangent point. The HI channel maps confirm no significant absorption features beyond 70 km s$^{-1}$. Furthermore, the $^{13}$CO clouds seen at 123.5 and 128.5 km s$^{-1}$ (tangent point) shows no corresponding HI absorption features (Figure 4 bottom panel green curve). Thus, we conclude that those clouds are located in the background and place the SNR near, at a distance of 4.7 ± 0.2 kpc.

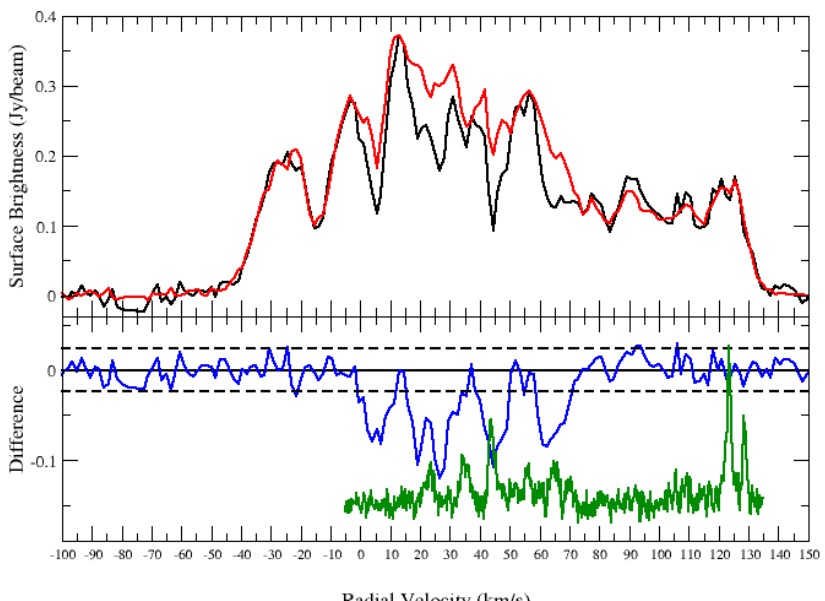

**Figure 4.** HI spectrum of the SNR candidate G18.760-0.072. Top panel: HI emission spectra (source: black and background: red). Bottom panel: source—background (blue). The dashed lines are ±2$\sigma$ noise level of the difference. The green curve in the lower panel is the $^{13}$CO emission spectrum.

### 3.3.2. SNR G31.299−0.493

G31.299−0.493 is ∼20″ (Figure 5a) in size. The 1.4 GHz MAGPIS image (Figure 5b) shows the diameter of the SNR to be ∼20″ and significantly larger than the synthesized beam of 6.2″ × 5.4″. This diameter is consistent with the shell-like structure seen in the 6 cm MAGPIS image (Figure 5c).

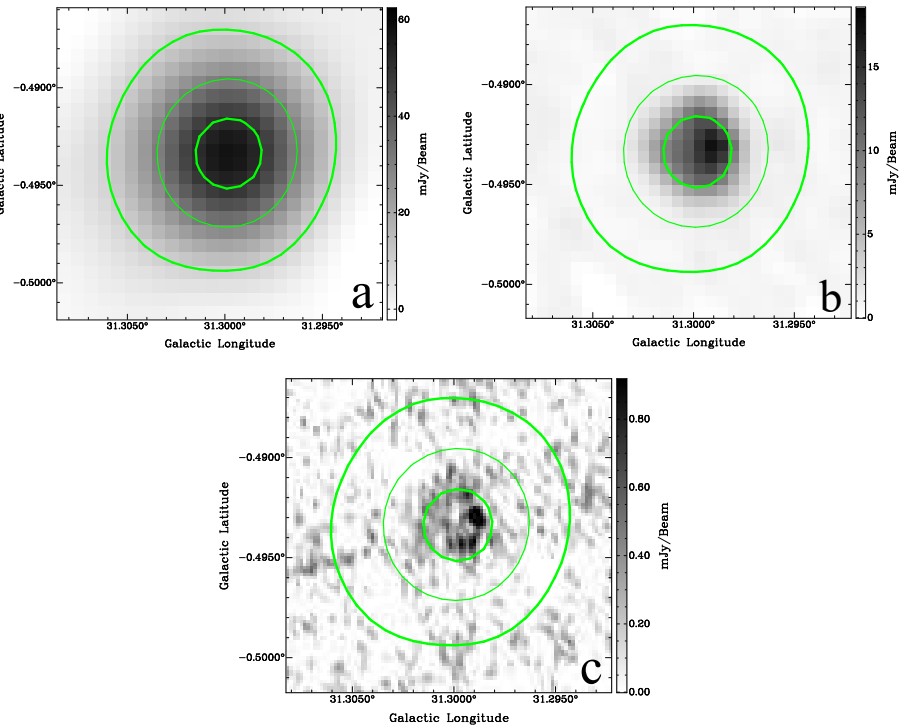

**Figure 5.** SNR G31.299−0.493 (**a**) THOR 1440 MHz continuum image; (**b**) MAGPIS 1.4 GHz image; (**c**) MAGPIS 6 cm image. The green contours are at 10, 30, and 50 mJy/beam from the THOR image.

The HI spectrum of G31.299−0.493 is shown in Figure 6. Significant HI absorption is detected in the positive velocity range above the $\pm 2\sigma$ noise level. With the $^{13}$CO emission coinciding with the HI absorption, the SNR is located at the distance corresponding to ∼85 km s$^{-1}$. The ∼85 km s$^{-1}$ corresponds to near and far distances of 5.0 and 9.2 kpc, respectively. However, the absorption is seen at <100 km s$^{-1}$ (tangent point $v_r \sim 110$ km s$^{-1}$), implying a location at the near kinematic distance of $5.0 \pm 0.3$ kpc.

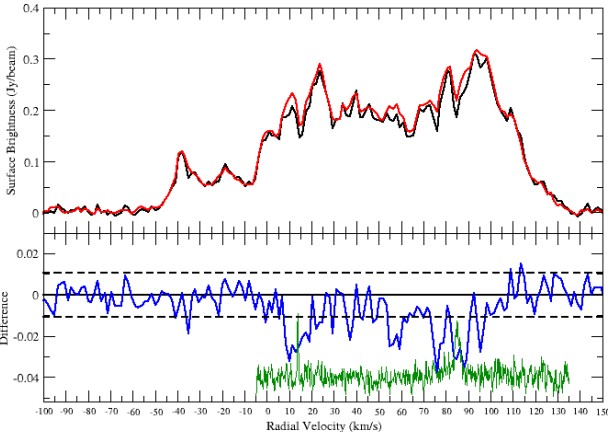

**Figure 6.** HI spectrum of the SNR G31.299−0.493. Top panel: HI emission spectra (source: black and background: red). Bottom panel: source—background (blue). The dashed lines are $\pm 2\sigma$ noise level of the difference. The green curve is the $^{13}$CO emission spectrum.

### 3.4. Radii of the 17 Galactic Non-Thermal Sources

With distances for the 17 galactic non-thermal sources, the radii can be calculated from the measured angular radii or upper limits. These are listed in Table 4. There is a clear division in the values of radius for the two SNRs ($\sim$1 pc) and for the 15 highly compact sources ($<$0.05 pc). As noted above, the highly compact sources are too small to be young SNRs. i.e., SNRs with typical expansion velocities of $v_{exp}$ $\sim$8000 km s$^{-1}$, would have ages less than 0.04 pc$/v_{exp}$ $\simeq$4.8 yr. The smaller ones with radii of $<$0.01 pc would have ages $\lesssim$1 yr. While it is possible for a SNR evolving in a medium with density of $> 10^6$ cm$^{-3}$ to have an age of $> 20$ yr, we believe it unlikely with these sources and thus categorize them as compact galactic non-thermal sources and not SNR candidates.

### 3.5. Spectral Indices

We determined the integrated flux densities for the 2 SNRs and the 15 sources with non-thermal spectra, using the TGSS, THOR, and CORNISH data. Where the CORNISH data are not available (brightness $<$ 2.5 mJy/beam), we use the MAGPIS 6 cm data. We measured the flux densities in a few different ways for each source and found the variations to be less than 10%. Therefore, we take the uncertainties of the flux densities for each continuum spectral window to be 10%. To model the spectrum, we used two models: a single powerlaw and a broken powerlaw with a low-frequency index and a high-frequency index. The best-fit spectral indices were calculated by minimizing $\chi^2$ in a fit to the flux densities vs. frequency. These values and the ones from [22] are given in Table 4.

The estimated spectral indices for the SNRs G18.760$-$0.072 and G31.299$-$0.493 are $-0.52 \pm 0.03$ and $-0.89 \pm 0.04$, respectively. The spectral index for G18.760$-$0.072 is typical for a SNR ($\alpha \sim -0.5$), and the spectral index for G31.299$-$0.493 is somewhat steeper. For the bright object G27.920+0.977 (peak intensity at $\sim$ 0.7 Jy/beam), the [22] spectral index and the one from this work agree.

Figure 7 shows the data and powerlaw fits from which the spectral indices were derived for the above 3 sources. For most (12 of 17) non-thermal sources, a broken powerlaw fits the data significantly better than a single powerlaw (Table 4).

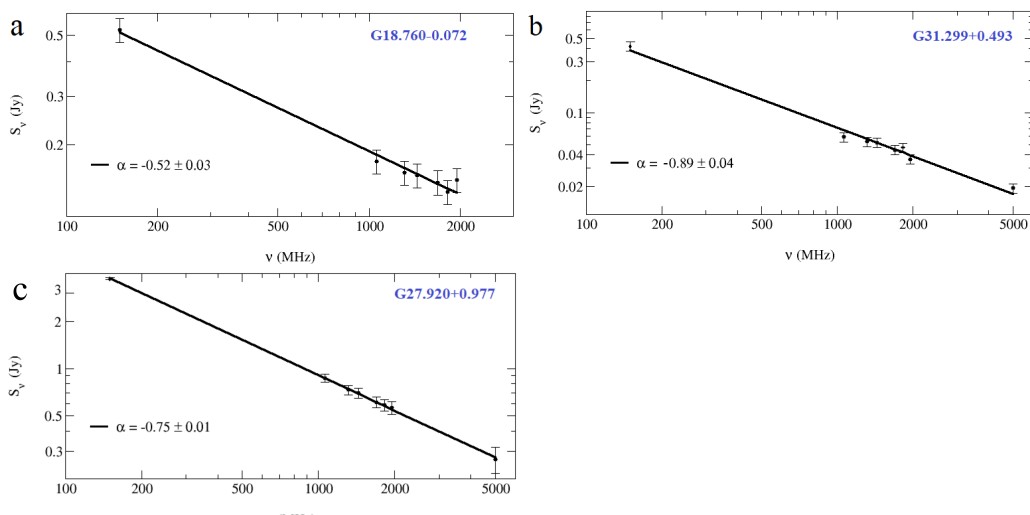

**Figure 7.** Continuum spectra for (**a**) SNR G18.760$-$0.072; (**b**) SNR G31.299$-$0.493; (**c**) G27.920+0.977; 150 MHz flux densities are obtained from the archival TGSS data; the flux densities between 1 and 2 GHz are from the THOR survey, and the 5 GHz data are from CORNISH and MAGPIS.

## 4. Discussion

### 4.1. The Two New SNRs

The galactic compact object (G31.299−0.493) shows a faint but ring-like morphology at 5 GHz, indicating the source to be a SNR. The compact object G18.76−0.072 was listed as a SNR candidate by [29]. The MAGPIS data confirm that it has extended emission (Figure 3).

In order to estimate the ages of these two SNRs, we use the models of [32,33], which calculate the ejecta dominated (ED) stage, and the transition ED-to-adiabatic and adiabatic stage. We assume uniform ISM $s = 0$, ejecta power law index $n = 7$, $0.5 \times 10^{51}$ erg explosion energy, ejected mass of $5 M_\odot$, and hydrogen number density $n_0 = 1$ cm$^{-3}$. For G31.299−0.493, the SNR is very young (10 yr). Lower ejecta mass or higher energy make the SNR age even smaller. With an associated $^{13}$CO feature, the SNR is likely expanding in a dense environment so that the age of the remnant would be larger, e.g., for $n_0 = 10^3$ cm$^{-3}$, the age is 57 yr. For G18.76−0.072, the SNR has an age of $\simeq 160$ yr. For a larger hydrogen number density ($n_0 = 100$ cm$^{-3}$), the age would be $\simeq 500$ yr. This places both of these SNRs in the category of the youngest several galactic SNRs.

### 4.2. The Other Galactic Non-Thermal Sources

We found 15 other non-thermal galactic sources (Sample 1 and Sample 2 in Table 4). For the 6 sources in Sample 1, two were found to be double radio sources in the higher resolution CORNISH and MAGPIS data. None of the nine sources in Sample 2 were resolved. All of these have angular radii < 1″, and small physical sizes < 0.05 pc (at their measured distances (Table 4)). SNRs of this size have ages $\lesssim$ 4–5 yr, and possibly much less because most sources are unresolved. It should be noted that the SN 1987A at 30 yr has a radius of ∼0.35 pc [34]. The 15 sources are significantly smaller than SN 1987A and the CSM densities a few orders of magnitude higher would be required for a comparable age. Thus, we consider them unlikely to be SNRs.

The 15 non-thermal galactic sources in Sample 1 and Sample 2 have broken power law spectra, except for the three sources G27.920+0.977, G36.204−0.342, and G44.324−0.730. The non-thermal emission from these galactic sources may be due to Wolf-Rayet systems, black holes, or novae. The double radio sources are separated by ∼15″ (∼0.7 pc) and are thus unlikely to be members of wide binary systems, although individually, they could be binary systems.

Ref. [35] presented the results of the Wolf-Rayet system WR 147, a radio source with thermal and non-thermal components. The source has a flux density of ∼28 mJy at 1.46 GHz and is nearly constant the final years of observations (their Table 4). However, the spectrum (their Figure 7) is predominantly thermal.

The candidate black hole X-ray binary MAXI J1631-472 has a spectral index ∼0.2–∼0.9, but the flux densities are in the order of a few mJy [36]. With the faintest source in our list having a flux density of ∼60 mJy, it is unlikely that the sources are Wolf-Rayet systems or black hole binaries.

The multi-frequency radio light curves of the nova V1500 Cyg presented by [37] show a flux density of ∼10 mJy at 2.7 GHz. While the radio emitting lifetime at 2.7 GHz is ∼3 yr, it may be longer at 1.4 Ghz. With a nova rate in the galaxy of ∼50/yr [37], we expect to find many radio novae. However, the galactic sources from this work are significantly brighter and are unlikely to be novae.

These non-thermal galactic sources may be candidates for colliding-wind binaries (CWB). The CWB observations presented by [38] have a flux density of ∼60 mJy at 2.28 GHz. Three sources from sample 1 (G35.351+0.239, G35.484+0.424, and G42.093−0.43) and six sources from sample 2 (G23.088+0.224, G36.204−0.342, G43.03−0.077, G44.324−0.73, G56.364+0.617, and G56.608−1.105) have comparable flux densities at 1.95 GHz.

*4.3. Implication for Number of Compact Galactic SNRs*

4.3.1. Angular Size Distribution of SNRs and THOR Sources

The angular radius distribution for the list of 294 galactic SNRs in the [1] catalogue is shown in Figure 8. We define effective angular radius as the average of the semi-major and semi-minor axes for each SNR. For sizes larger than 10′, the angular radius distribution is fit by a power law with slope ≃ −1. Below ∼5′, there is a decrease in the number of SNRs, with only two SNRs with angular radius less than 1′. Identified SNRs are clearly deficient in number below ∼5′ in radius.

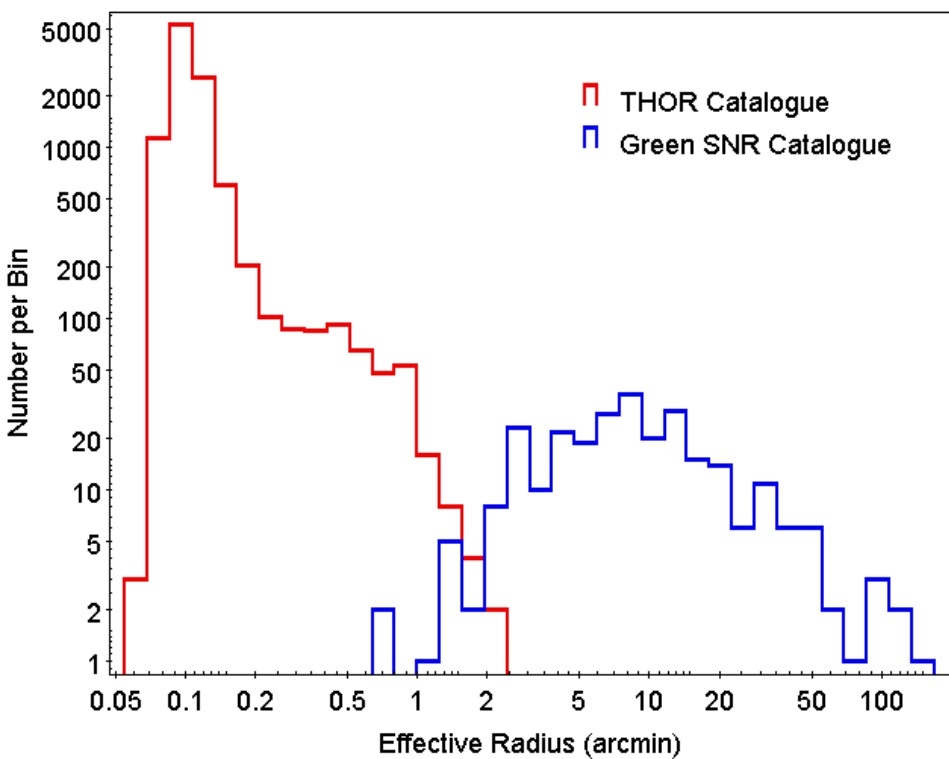

**Figure 8.** Distribution of angular radii of sources from the catalogue of [22] (red) and Green's SNR catalogue (blue).

There is a wide range in known distances to SNRs, from a few hundred pc to ∼20 kpc. For a mean distance of ∼6 kpc, a 5′ radius corresponds to a physical radius of 8.7 pc. This is quite large, corresponding to a Sedov age of ≃4,000 yr for an explosion energy of $10^{51}$ erg and ISM density of 1 cm$^{-3}$. Explosion energies and ISM densities for galactic SNRs vary by an order of magnitude [39]; thus, the corresponding ages for this radius can vary by an order of magnitude (i.e., ∼400 to ∼40,000 yr).

The radio brightness of a SNR is thought to peak at the end of the ejecta dominated stage [40]. The ejecta dominated stages ends gradually between the time when the reverse shock hits the ejecta core, at ∼300 yr and radius ∼2.5 pc, and when the reverse shock hits the centre, at age ∼1500 yr and radius ∼5.5 pc [32,41]. Thus, SNRs are expected to be radio bright at size ∼3 pc, which corresponds to an angular radius of ∼1.5″ at a distance of 6 kpc and ∼0.5′ at distance of 20 kpc. This is consistent with the decrease in SNR number for the small angular radii seen in Figure 8.

The angular radius distribution[6] for the objects in the THOR catalogue is compared with that of SNRs in Figure 8. The THOR sources fill in much of the SNR deficit region below 3′ of the SNR distribution. The THOR size distribution has a peak at 0.1′, with a drop-off at smaller sizes likely caused by large uncertainties of beam-corrected sizes. The observed THOR normalization is higher than the SNR normalization because the THOR catalogue includes many types of objects that are much more numerous than SNRs, mainly extragalactic sources.

From the observed SNR angular radius distribution, the angular size region of most interest for searching for new SNRs is the range below $3'$. This corresponds to SNR ages of $\lesssim 1000$ yr. The number of missing SNRs in the radius distribution of known SNRs (Figure 8) depends on the model of SNR radio emission vs. size and age, the details of which are not the focus of the current study. However, we make a simple estimate, assuming the radio flux density is constant during this part of the SNR evolution. Then, the number of SNRs in this size range is proportional to the time span of the corresponding age range. The radius evolves as a changing power-law between the ejecta-dominated value of $R \propto t^{(n-3)/(n-s)}$ [42], with $n$ and $s$ defined above, and the adiabatic value $R \propto t^{2/5}$. A typical $n = 10$, $s = 0$ (uniform external medium) gives an ejecta-dominated evolution of $R \propto t^{7/10}$, and a typical $n = 10$, $s = 2$ (wind external medium) gives an ejecta-dominated evolution of $R \propto t^{7/8}$.

The faster evolution of $R$ for young SNRs is an important ingredient in the explanation of why there are fewer SNRs at smaller angular radii. For ejecta-dominated SNRs, $dt/dR \propto R^{3/7}$ for $s = 0$ and $dt/dR \propto R^{1/7}$ for $s = 2$. As we use logarithmic sized bins in Figure 8 ($dR \propto R$), the number of expected SNRs per bin increases as $N \propto R^{10/7}$ ($s = 0$) and $N \propto R^{8/7}$ ($s = 2$). If there is no angular size selection in the range of interest, the number of detected SNRs should increase, as just noted.

For the later SNR adiabatic evolution, one finds $N \propto R^{5/2}$ (for logarithmic bins). The fact that the observed distribution is much flatter than $R^{5/2}$ for a large R is due to the strong selection effect against larger SNRs. The decreasing radio brightness and difficulty of distinguishing SNRs from background radio emission in the galaxy are both important in selecting against larger SNRs.

For small SNRs, the observed slope in Figure 8 for angular sizes less than $3'$ is $\sim 2.3$. The smallest and youngest SNRs ($\lesssim 1000$ yr) are still in the ejecta-dominated phase and should have evolution as noted above, which yields between $N \propto R^{10/7}$ and $N \propto R^{8/7}$. The comparison with the observed slope from Figure 8 shows that there is a significant angular size-dependent selection effect for young SNRs. i.e., the observed number of small SNRs is below the expected number by $\sim$30–40 if the radio brightness is constant (as we assumed) or by a smaller number ($\sim$15–20) if the radio brightness increases with SNR age (and angular size).

An alternatice way to verify the missing number of young SNRs is by use of the SN birth rate of the galaxy of 1/(40 yr). This yields an expected number of 25 SNRs younger than 1000 yr. This is significantly more than the known number of SNRs younger than 1000 yr of $\sim$6 to 10. The main uncertainty here is determination of ages for SNRs—this problem has been studied for the subset of SNRs with X-ray spectra [39], which allow for age determination. In summary, the expected number of previously undiscovered SNRs less than 1000 yr old is $\sim$15 to 20.

### 4.3.2. Flux Density Distribution of SNRs and THOR Sources

Figure 9 compares the integrated flux densities of the Green's catalogue SNRs and of THOR catalogue sources. The SNRs have higher flux densities, but there is considerable overlap with the range of flux densities of THOR sources. The extrapolation of the SNR flux densities to smaller angular radii overlaps the region where THOR sources are located, although the extension of the lower envelope of SNR flux densities is below the region occupied by THOR sources. This implies that $\sim$half of the small angular radius SNRs should be bright enough to be detectable by THOR.

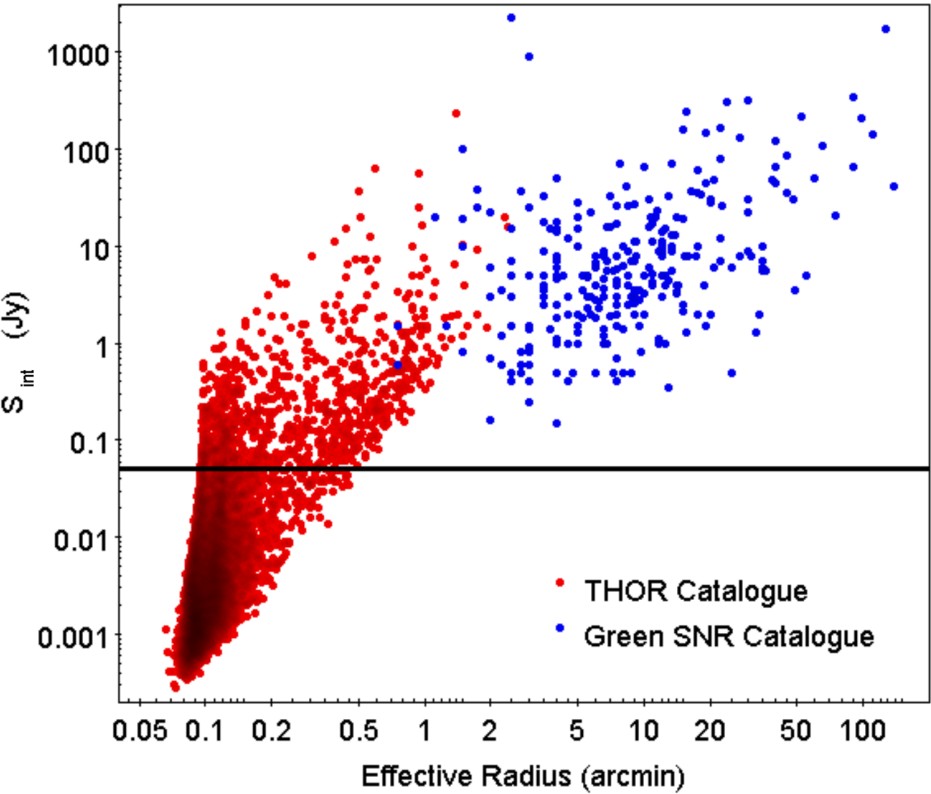

**Figure 9.** Distribution of integrated flux densities at 1.4 GHz of sources from the catalogue of [22] (red) and Green's SNR catalogue (blue). Sources above 50 mJy (horizontal line) were chosen to obtain HI spectra.

### 4.3.3. Expected vs. Detected Number of New THOR SNRs

From Section 4.3.1, we estimated that ∼15 to 20 undiscovered young (<1000 yr) SNRs exist in the galaxy. Scaled down to the THOR fraction of the galactic plane, the expected number in THOR is ∼3 to 4. We found two new small/young SNRs in the search of the THOR data i.e., a significant fraction (∼1/2) of the expected number. From Section 4.3.2 and Figure 9, we estimated that the THOR survey is sensitive to ∼1/2 of the small angular size (< 3′) SNRs, which is consistent with the above.

### 4.4. *General Remarks*

One issue is whether the observational data can provide evidence on the type of observed SNRs or identity of the SN progenitors. With X-ray data, it is possible to deduce the energy and ejected mass of supernova resulting in the remnant and thus probable SN type (e.g., [39]). However, with radio data (these new SNRs have no X-ray data), it so far has not been possible to do so. Here, we ran SNR evolutionary models to match measured radii, with the assumption of ejected mass of 5$M_\odot$. The age differences between models for ejecta masses 1.4 and 5$M_\odot$ for SNRs with small radii are small (10–15%), calculated by running the models.

Another question is how numerical simulations relate to observations of SNRs. Virtually all simulations have been carried out with the aim of understanding individual SNRs. Examples of recent three-dimensional simulations are presented by [43–45], and earlier simulations can be found in the references to those papers. These simulations have different goals. Ref. [43] matched the high energy emission (few keV X-rays to ∼10 MeV gamma-rays) of SN 1987A in order to test different explosion types against observations. Ref. [45] compared the X-ray emission properties of Type Ia SNRs produced by Chandrasekhar and Sub-Chandrasekhar mass progenitors to see if the X-ray properties of SNRs can be used to discern the Ia progenitor white dwarf mass. Ref. [44] simulated

explosions with 1.4 and 10 $M_\odot$ ejecta and a set of explosion energies and ambient uniform densities and then coupled the hydrodynamic solution with a nonlinear diffusive shock acceleration code to calculate the particle acceleration and radio emission. They found that the range of input energies, ambient densities, and ages can produce a spread in radio surface brightness vs. SNR diameter consistent with observed SNRs. These simulations have promise in constraining SNR properties, including cosmic ray injection parameter and magnetic field damping parameter (their Figure 4) but are not sensitive to explosion mass. Similar simulations, which have not been conducted yet to our knowledge, could be used to explore radio lifetimes of old SNRs, which are useful for estimating number of old SNRs or for investigating the early radio evolution of SNRs, which would be useful in predicting the number of young SNRs.

## 5. Conclusions

The currently known set of SNRs is deficient in small and young SNRs as well as in large and old SNRs. The THOR survey covers the region of parameter space where one expects to find small/young SNRs, and thus is a useful data set in which to search for new SNRs. HI data are indispensable for the identification of compact galactic non-thermal sources buried in a much larger set of extragalactic sources.

Using the THOR source catalogue [22], we carried out a search for compact candidate SNRs using the SNR criteria in Section 1. We found 350 compact objects with non-thermal spectral indices and constructed HI absorption spectra to determine whether they are galactic sources. HI absorption distances to the galactic sources were determined, which allows for the conversion of angular radii to physical radii. Of the galactic non-thermal compact sources, 17 that were not associated with HII regions were identified. The basic parameters of 17 objects are given in Table 4.

We found two SNRs: the new SNR (G31.299−0.493) and the candidate G18.760−0.072 [29] confirmed as SNRs. The other 15 objects are too compact (<0.05 pc) to be SNRs. The number of new SNRs is a significant fraction ($\sim$1/2) of the expected number of missing young SNRs, given the fraction of the galaxy that is covered by the THOR survey.

**Supplementary Materials:** The supplementary material is available online at https://www.mdpi.com/article/10.3390/universe7090338/s1.

**Author Contributions:** Conceptualization, J.S. and D.L.; formal analysis, S.R., D.L. and J.S.; resources, J.S.; data curation, S.R.; writing—original draft preparation, S.R. and D.L.; writing—review and editing, J.S. All authors have read and agreed to the published version of the manuscript.

**Funding:** This work was supported by a grant from the Natural Sciences and Engineering Research Council of Canada.

**Institutional Review Board Statement:** Not applicable.

**Informed Consent Statement:** Not applicable.

**Data Availability Statement:** This research has made use of the THOR data provided by the THOR team. The data were accessed on September 2019. The THOR data products are available at the website https://www2.mpia-hd.mpg.de/thor/Overview.html. The TGSS data were accessed on August 2020 and the data products are available at http://tgssadr.strw.leidenuniv.nl/doku.php. The SPIDX progressive survey was accessed on 3 March 2020 and is available at http://tgssadr.strw.leidenuniv.nl/hips_spidx/. The CORNISH survey data can be found at https://cornish.leeds.ac.uk/public/index.php and accessed on 8 October 2019. The MAGPIS, GLIMPSE and MIPSGAL data accessed on 16 December 2020 and the data products are available at https://third.ucllnl.org/gps/index.html.

**Acknowledgments:** We thank John Dickey for the helpful comments.

**Conflicts of Interest:** The authors declare no conflict of interest.

## Appendix A. Distance Estimations of the 15 Non-Thermal Sources

The HI spectra, analyses, and the distance estimations of Sample 1 and Sample 2 (six from sample 1 and nine from sample 2) are given below. See Section 3.3 for the details.

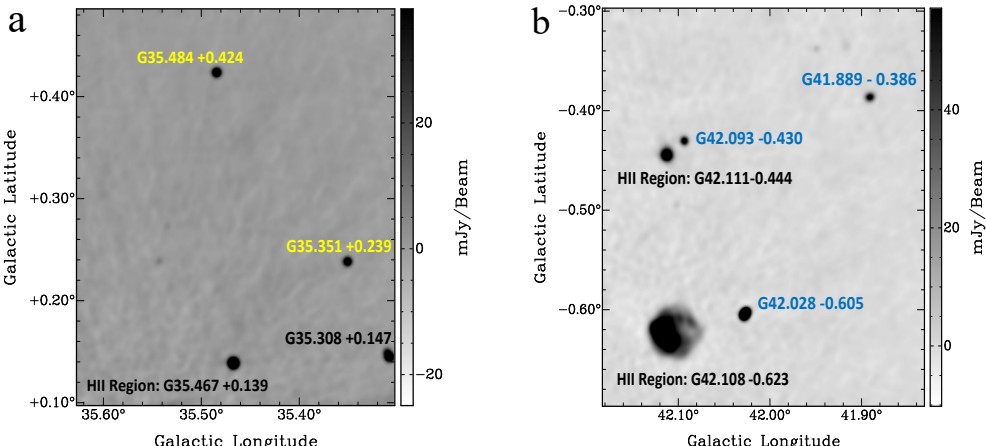

**Figure A1.** (**a**) Non-thermal sources G35.351+0.239 and G35.484+0.424, the IR quiet object G35.308+0.147, and the HII region G35.467+0.139 1440 MHz THOR continuum image; (**b**) Non-thermal sources G41.889−0.386, G42.028−0.605, and G42.093−0.430; HII regions G42.108−0.623 and G42.093−0.430 1440 MHz THOR continuum image. The HI spectra of the HII regions and IR-quiet object were used for comparison.

*Appendix A.1. Sample 1 Sources*

Appendix A.1.1. G27.920+0.977

The bright source (with a peak intensity of ∼0.7 Jy/beam) G27.920+0.977 shows absorption up to the tangent point (112 km s$^{-1}$) (Figure A2A). The absorption features seen at −28 and −40 km s$^{-1}$ are features likely due to HI clouds with peculiar velocities. The morphology of the absorption features are inconsistent with the 1440 MHz radio continuum morphology, and we conclude that no absorption is seen at the negative velocity range. The SNR candidate is located between the tangent point distance (7.3 kpc) and the far side of the solar circle (14.7 kpc).

There are no bright sources near the SNR candidate (<30′) for comparison. However, examining the spectrum and HI channel maps, it is seen that there is no absorption between 47 and 55 km s$^{-1}$. Therefore, the SNR candidate is likely located at the distance of 11.3 ± 0.3 kpc, which corresponds to the velocity of 55 km s$^{-1}$.

Appendix A.1.2. G35.351+0.239

From the HI spectrum (Figure A2B) of G35.351+0.239 (Figure A1a), we see absorption in the negative velocity range. However, the HI channel map investigation confirms that the absorption features seen in the negative velocity range are coincidental. The IR quiet object G35.308+0.147 (see Figure A1 for its location) shows absorption in the negative velocity range (absorption features seen between −20 and −30 km s$^{-1}$ and −50 km s$^{-1}$, Figure A2D).

The nearby HII region G35.468+0.139 (Figure A1a) has a V$_{LSR}$ = 80.9 ± 05 km s$^{-1}$ [25]. Ref. [25] presented the tangent point velocity as 87.8 km s$^{-1}$, and because the velocity of the source is within 10 km s$^{-1}$, the kinematic distance ambiguity was not resolved. i.e., the estimated distance is consistent with either the near distance of 5.6 kpc or the far distance of 8.2 kpc. From the HI emission spectra (Figure A2E), it is seen that the tangent point velocity is ∼98 km s$^{-1}$. This velocity is consistent with the [5] rotation curve, which yields ∼94.7 km s$^{-1}$. The HII region shows absorption at 88 km s$^{-1}$ and, therefore, is located nearby, at a distance of 5.5 kpc.

The IR quiet object shows absorption up to ~98 km s$^{-1}$ (tangent point). Neither the SNR candidate nor the HII region show absorption up to the tangent point. The absorption feature seen at ~90 km s$^{-1}$ is coincidental; however, the real absorption features are seen at 80.5 km s$^{-1}$. Thus, we place G35.351+0.239 nearby, at a distance of 4.9 ± 0.4 kpc.

### Appendix A.1.3. G35.484+0.424

G35.484+0.424 (Figure A1a) shows absorption up to the tangent point, ~95.5 km s$^{-1}$ (Figure A2C), and is located beyond the tangent point. The HI channel map investigation shows that the absorption features seen at the negative velocity range are coincidental. We place the SNR candidate between the tangent point (6.8 kpc) and the solar circle (13.5 kpc).

### Appendix A.1.4. G41.889−0.386

G41.889−0.386 (Figure A1b) shows absorption up to the tangent point (Figure A2F). The HI channel map investigation shows no absorption in the negative velocity range. The SNR candidate is located between the tangent point (6.2 kpc) and the solar circle (12.4 kpc).

The HII regions G42.111−0.444 and G42.108−0.623 (Figure A1b) are located at ~14′ and ~20′ from the source. G41.889−0.386 does not show absorption at 55 km s$^{-1}$. However, the HII region G42.111−0.444 shows clear absorption at this velocity. Therefore, we estimate the distance to be 8.9 ± 0.4 kpc.

### Appendix A.1.5. G42.028−0.605

G42.028−0.605 is located near the HII region G42.108−0.623 (Figure A1b). The non-thermal source shows absorption in the positive velocity range up to the tangent point (Figure A2G). From the HI channel maps, the absorption seen at −10 km s$^{-1}$ is found to be coincidental. Therefore, the SNR candidate is located between the tangent point (6.2 kpc) and the far side of the Solar circle (12.3 kpc).

As the HII region is nearby, we compare both the spectra and the HI channel map absorption features. The HII region spectrum (Figure A2I) and the HI channel map investigation show absorption up to the tangent point, indicating that the HII region is located beyond the tangent point. Ref. [24] presents the HII region $V_{LSR} = 66.0 ± 0.6$ km s$^{-1}$, which is consistent with the absorption features. This corresponds to a distance of 8.9 kpc. However, strong absorption is found for the SNR candidate between ~20 and ~30 km s$^{-1}$, while none is found for the HII region. Therefore, the lower limit for the SNR distance estimation is the distance of 11 kpc, which corresponds to the velocity of 20.5 km s$^{-1}$. Taking the average, we place the candidate at a distance of 11.7 ± 1.1 kpc.

### Appendix A.1.6. G42.093−0.430

The non-thermal source G42.093−0.430 is located ~1.2′ from the HII region G42.111−0.444 (Figure A1b). The HII region shows absorption up to the tangent point and has a $V_{LSR} = 53.4 ± 0.6$ km s$^{-1}$ [25]. The HII region is located at a distance of 7.7 kpc.

G42.093−0.430 does not show absorption up to the tangent point (76.8 km s$^{-1}$) (Figure A2J). The HI channel map investigation shows clear absorption for the HII region at 79 km s$^{-1}$ but not for G42.093−0.430. Absorption for the non-thermal source G42.093−0.430 is seen at 65.5 km s$^{-1}$, which corresponds to the near distance of 4.3 ± 0.5 kpc.

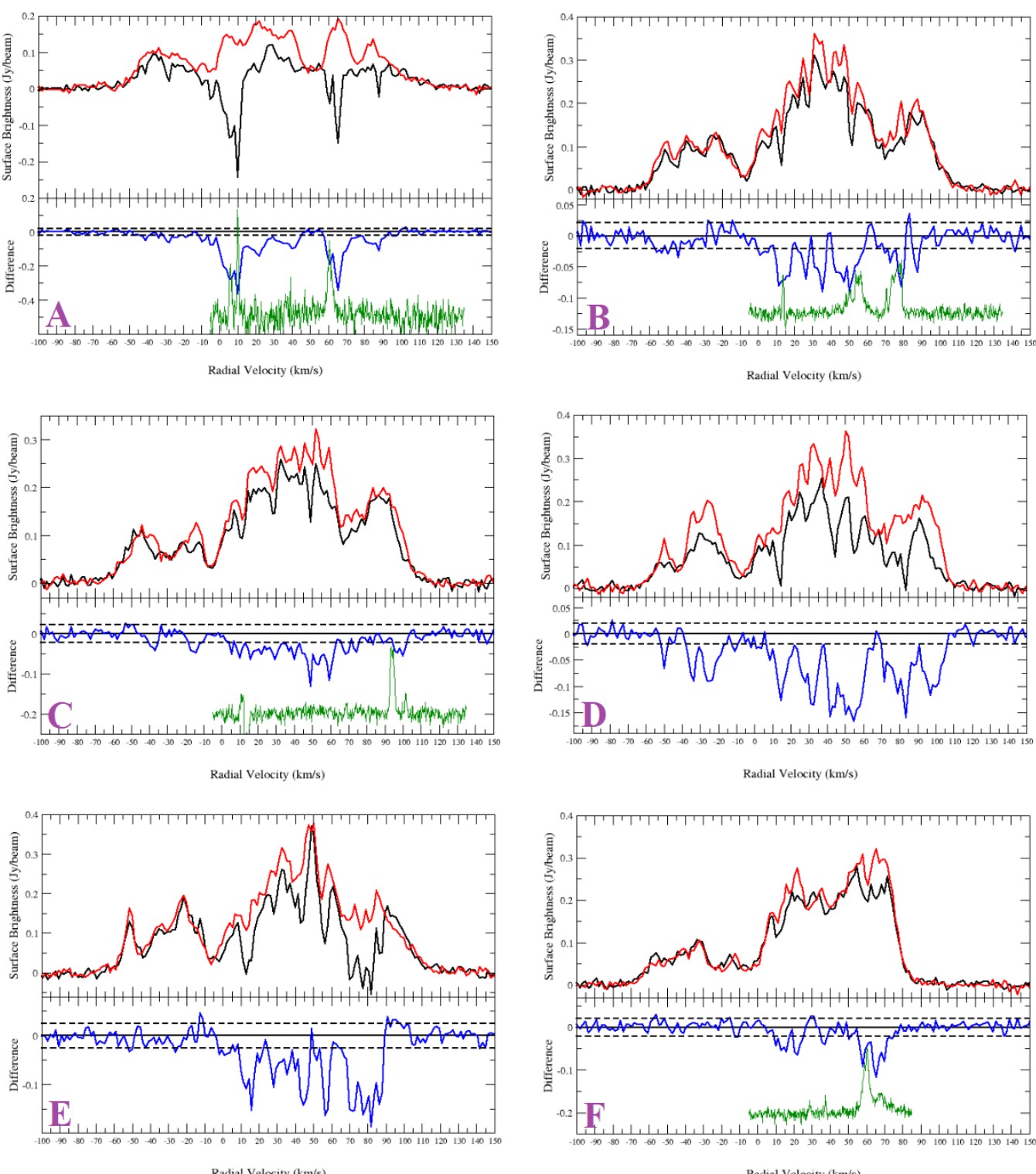

**Figure A2.** *Cont.*

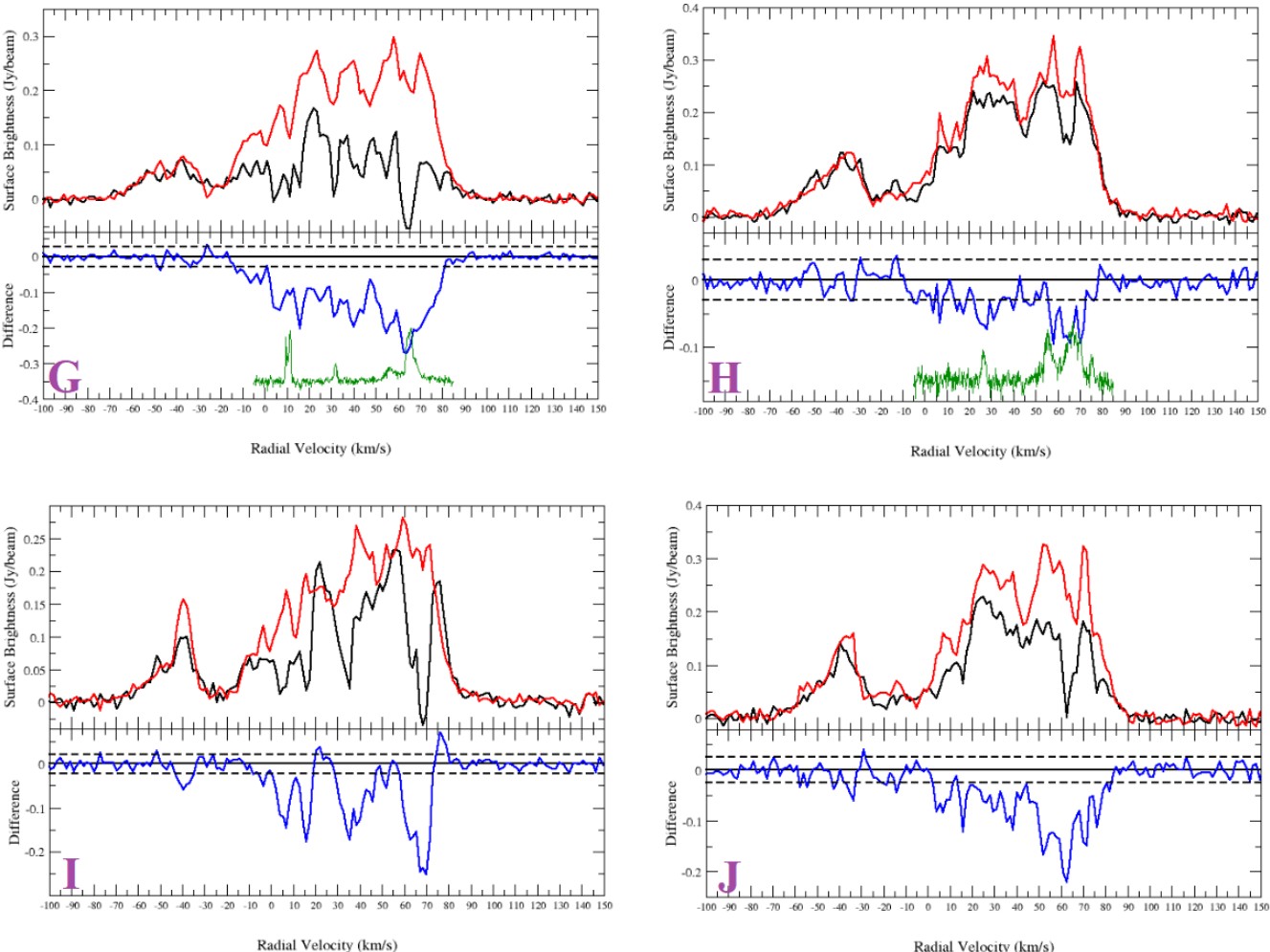

**Figure A2.** HI spectra of the sources from Sample 1 and nearby objects. Non-thermal sources: (**A**) G27.920+0.977; (**B**) G35.351+0.239; (**C**) G35.484+0.424; (**F**) G41.889−0.386; (**G**) G42.028−0.605; (**H**) G42.093−0.430. HII Regions (comparison sources): (**E**) G35.467+0.139; (**I**) G42.108+0.623; (**J**) G42.111−0.444. IR-quiet sources (comparison source): (**D**) G35.308+0.147. Top half of panel: HI emission spectra (source: black and background: red). Bottom half of panel: source—background (difference: blue). The dashed lines are ±2σ noise level of the difference. The green curves plotted in the lower halves of the panels are the $^{13}$CO emission spectra.

*Appendix A.2. Sample 2 Sources*

Sample 2 sources do not have sources nearby to constrain their distances. The HI channel map analyses shows very little evidence of lack of absorption. For this reason, we give a lower limit distance for most of the sources.

Appendix A.2.1. G23.088+0.224

The HI spectra and the channel maps show absorption up to the tangent point (Figure A3K). With no absorption features seen in the negative velocity range, we place the source G23.088+0.224 beyond the tangent point of 7.6 kpc.

Appendix A.2.2. G24.430−1.017

G24.430−1.017 shows absorption features up to the tangent point (Figure A3L). HI channel maps do not show conclusive evidence of absorption features. The source is located beyond the tangent point of 7.6 kpc.

### Appendix A.2.3. G34.421−1.031

G34.421−1.031 does not show absorption up to the tangent point ($\sim$ 95 km s$^{-1}$). The HI channel maps show absorption at 85 km s$^{-1}$ (Figure A3M). We place the source at a distance of 5.1 ± 0.4.

### Appendix A.2.4. G36.204−0.342

The HI spectrum of G36.204−0.342 shows clear absorption features up to the tangent point and none in the negative velocity range (Figure A3N). The source is located beyond the tangent point of 6.9 kpc.

### Appendix A.2.5. G43.030−0.077

G43.030−0.077 shows absorption features up to the tangent point and, therefore, is located beyond the tangent point (Figure A3O). However, no absorption is seen at 37 km s$^{-1}$. The source is located at the far distance just within the corresponding distance of 9.8 ± 1.4 kpc.

### Appendix A.2.6. G44.324−0.730

The source G44.324−0.730 shows absorption in the positive velocity range and up to the tangent point (Figure A3P). The lower limit estimate of the distance is 6.1 kpc.

### Appendix A.2.7. G56.364+0.617

The HI channel maps and spectra confirm absorption features up to the tangent point for the source G56.364+0.617 (Figure A3Q). The absorption features seen in the negative velocity range are coincidental false features. The source lies beyond the tangent point distance of 4.6 kpc.

### Appendix A.2.8. G56.608−1.105

G56.608−1.105 shows absorption up to the tangent point (Figure A3R) and is located at least at a distance of 4.6 kpc.

### Appendix A.2.9. G64.019−0.846

G64.019−0.846 Shows absorption in the whole positive velocity range (Figure A3S) . The lower limit estimation of the distance is 3.6 kpc (tangent point).

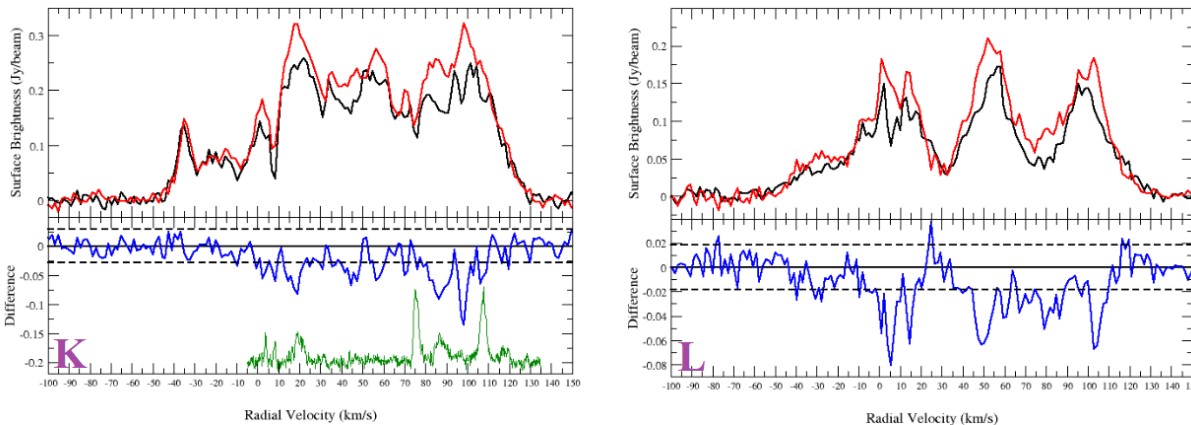

**Figure A3.** *Cont.*

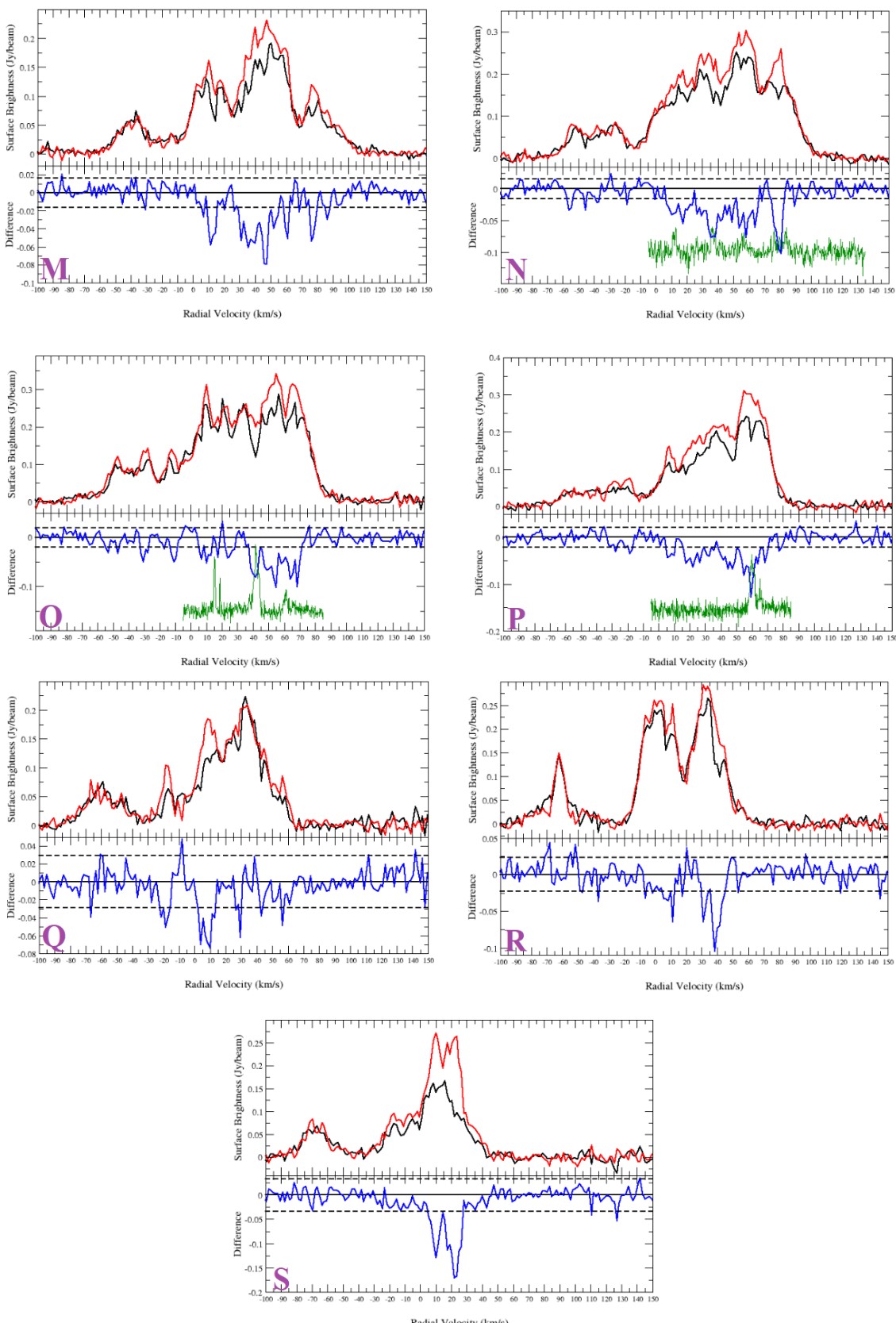

**Figure A3.** HI spectra of the sources from Sample 2. (**K**) G23.088+0.224; (**L**) G24.430−1.017; (**M**) G34.421−1.031; (**N**) G36.204−0.342; (**O**) G43.030−0.077; (**P**) G44.324−0.730; (**Q**) G56.364+0.617; (**R**) G56.608−1.105; (**S**) G64.019−0.846. Top half of panel: HI emission spectra (source: black and background: red). Bottom half of panel: source—background (difference: blue). The dashed lines are $\pm 2\sigma$ noise level of the difference. The green curves plotted in the lower halves of the panels are the $^{13}$CO emission spectra.

## Notes

1  Of the 294 known SNRs, there are 55 SNRs in the area covered by THOR survey.

2  THOR covers $\sim 1/4$ of the area of the galactic disc for a galactic radius of 16 kpc.

3  http://third.ucllnl.org/gps, accessed on 16 December 2020.

4  The THOR survey objects are in the first quadrant of the galaxy, so absorption from the far side of the galaxy outside the solar circle is at negative velocities.

5  The lack of absorption features at the negative velocity range does not necessarily mean that they are galactic sources due to the low abundance of cold HI gas towards the edge of the galaxy. However, a comparison with the spectrum a nearby known galactic (e.g., HII region) or extragalactic objects yields evidence of absorbing HI gas in the outer galaxy, and a more definite conclusion on the distance and galactic or extragalactic nature of the candidate.

6  We tested two definitions of effective angular radius. For the first, we used the radius that gives the same circular area ($\pi r^2$) as the island area of each source in the catalogue. The threshold of defining an island for each source was $2.6\sigma$ above background. This definition gives a smallest size, which is 1 pixel area and does not account for source smearing by the beam area. For the second, we used a radius determined by the ratio $C =$ (integrated flux density)/(peak flux density) given by $\theta = \theta_{beam} \times \sqrt{(C)}$, which is valid for a Gaussian-shaped source. When we compare radius distributions for the two definitions, we find the first definition results in a peak at near 25'' caused by the beam, so we chose to use the second definition.

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
