# Peer review of "Search for and Identification of Young Compact Galactic Supernova Remnants Using THOR"

_universe, doi:10.3390/universe7090338_

Round 1
Reviewer 1 Report
This is a referee report for the paper:
Search for and Identification of Compact Galactic SNRs Using THOR
by S. Ranasinghe*, D. Leahy and J. Stil
This is a nice paper, where the authors have used the THOR radio survey to look for new SNRs, and find 2 SNRs, one of which is new and one which had been suggested previously as a candidate.
I liked the paper, although I am going to suggest to the authors that they change the order of the presentation below to make it much more readable. I don't think I have any major complaints about the paper, but I do have several comments and suggestions that I would like the authors to address before the paper can be accepted to the journal.
Title: Spell out "Supernova Remnants" in the title.
Introduction: A little more description of the role of other radio surveys, and what SNRs they have discovered, is required. It is for example known that low frequency surveys have found a few radio remnants not discovered before.
Line 26: lack of number of observed SNRs --> remove 'number of'.
Line 32: "For catalogued SNRs with distance estimates", do you mean catalogued in general, or from the THOR survey? I think it means in general, in which case a reference is needed to which catalog you are referring to and where these numbers come from.
Line 43: I presume you mean an index that is something like -0.25 or smaller. You would not allow something like -0.1 because that would be too close to thermal bremsstrahlung. Actually I see that this is addressed later in Line 93, but it should be mentioned here.
Line 101: " radio sources that have a double-lobe morphology" - In principle a SNR could have such a morphology. Especially if it exploded inside a bipolar bubble. SN 1987A comes immediately to mind. There wouldn't be too many, but there could be the odd one. Did the authors look into size for example, or any other parameter before identifying these as extragalactic?
Line 108: "150 objects show absorption.." -- How many of these were coincident with the 77 that showed double lobe morphology? Were these two sets mutually exclusive? It seems like that, but some mention must be made.
Line 168: "This leaves the two largest angular radius ones.." - Please mention by number so its obvious which ones you are referring to , i.e. objects 1 and 2.
Line 209: I am not so convinced by this argument. What if they were evolving in a very high density? Wind densities close in to a SN can be of order 10^6 to 10^7 cm^-3. That would make them much older. I don't expect too many. However young SNe evolving in a high density region could be a few tens of years old with this size. It is possible that if you cannot resolve them you cannot do much more with them, but that does not mean they cannot be SNR candidates. At least I do not think this argument is sufficient.
Line 225: Remove "[20] estimated the spectral index using the THOR data." or put that sentence somewhere else. It does not connect with the rest of this paragraph which is dealing with the authors own estimation of the remnant.
Line 247: "the age is 57 yr" --> add 'is'
Line 256-257 : "small sizes make them highly unlikely to be SNRs" --> unless of course the density is really high, which is possible. SN 1987A at 30 years was still confined to the equatorial ring, with a radius 0.2pc. If the density were 10-100 times larger then the size would be comparable to the sizes seen here.
Section 4.3 - I am going to suggest that some of these predictions come right at the beginning, after talking about THOR. It would be much better to have a discussion of how many remnants could be expected in THOR and how many remnants the authors should see, and then go on to actually searching for remnants. At the moment, this sub-section appears to not tie in with the rest of this section, and it seems this discussion may be better served in the beginning. I hope the authors will consider this.
Overall the paper is reasonably well written, without any major negatives. I feel that the authors need to address the above comments before the paper can be accepted.
Reviewer 2 Report
The manuscript elegantly presents state-of-art results meriting publication in Universe. I would suggest the authors to further discuss the following minor points: (1) is there any conclusions which could be provided from the data regarding to the type of observed supernova remnants, and, therefore, to the identity (mass) of the supernova progenitors ? (2) The paper would benefit a discussion of the observational results in the context of numerical simulations of supernova remnants.
